

# Thermal conditions on the coast of Labrador during the late 18th century

Garima Singh[1], Rajmund Przybylak[1,2], Przemysław Wyszyński[1,2], Andrzej Araźny[1,2], Konrad Chmist[1]

[1] Faculty of Earth Sciences and Spatial Management, Nicolaus Copernicus University in Toruń, Poland
[2] Centre for Climate Change Research, Nicolaus Copernicus University in Toruń, Poland

Corresponding author: Garima Singh, garima.singh@doktorant.umk.pl

**Abstract.** In this article, we present research results on the air temperature changes on the Labrador coast at the end of the 18th century (1771–87). This important climatic variable was studied on the
basis of valuable instrumental meteorological observations made by Moravian missionaries. The data were taken from meteorological registers available in three major archives: the Moravian Archives in Herrnhut (Germany) and the Moravian Archives at Muswell Hill and the Archives of the Royal Society, both located in London (United Kingdom). The series of meteorological observations we used in the paper are the oldest and longest long-term meteorological observations available not only for
Labrador, but for anywhere in the entire Arctic. Moravian missionaries measured not only air temperature (analysed here) but also atmospheric pressure and wind (force and directions) two, three or even four times a day. These unique data allow us to better understand the climate variability and trends in the region during the study's historical period. We have analysed sub-daily air temperature readings from three sites: Okak (1776–87), Nain (1771–86) and Hopedale (1782–86). The data were
converted into more relatable present-day units and have undergone rigorous quality control. Original mean daily air temperature data calculated using different numbers of measurements per day were corrected to the "real" mean daily values. The corrected values were subsequently used for statistical analysis. The historical temperatures documented during our specified study period along the Labrador coast were compared with those experienced today. The analysis shows a significant warming from
historical to present times. Historical data from Nain, Okak, and Hopedale, representing different periods were, on average, about 0.5 to 2.3 °C colder than in the modern period of 1990 to 2020, especially in winter and autumn. Most monthly mean air temperatures in historical times lie within two standard deviations of the modern mean. The frequency of temperature occurrence in one-day intervals suggests a shift towards more stable and less extreme temperature distributions in contemporary times,
implying substantial changes in climate patterns over time in this region. The continentality of the Labrador climate, as well as year-to-year variability of mean monthly temperatures, were greater in historical times than at present.

Keywords: Labrador, climate change, air temperature, historical climatology, instrumental
observations, Moravian Church

## 1 Introduction

Investigating historical climate changes, especially for the last millennium, is crucial for understanding
past climate systems and, as a result, for constructing more reliable scenarios of future climate. Reconstructing the features of the climate in the last millennium, including the range of its natural changes, is particularly important for the Arctic, where recently observed changes in climate and environment have been greatest (Walsh et al., 2011; Jeffries et al., 2013; IPCC 2021). Improved knowledge about the range and rate of climate fluctuations in the last millennium in the Arctic, in
particular prior to the start of regular instrumental observations covering much of the area (mid-20th century, see Przybylak, 2000, 2016; Przybylak et al., 2024), can help in estimating the magnitude of



anthropogenic influences on Arctic and global climate. Therefore, there is an urgent need to collect and analyse any meteorological data for this period for every area of the world, particularly the Arctic.

In this paper, we present the results of our investigation of thermal conditions on the coast of Labrador (Canada) in the late 18th century based on meteorological observations made by missionaries of the Moravian Church (also known as the Unitas Fratrum or the Moravian Brethren) at three sites: Nain, Okak (also spelt Okkak) and Hopedale (Ger. *Hoffenthal*). The first mission on the Labrador coast was established in Nain in August 1771, and a little later, on 15th October, meteorological observations began. Thus, this date designates the beginning of instrumental

meteorological observations, which at the beginning were done by Christopher Brasen, the leader of the missionary settlement, until his tragic death on 15[th] September 1774. Encouraged by their early success, the Moravians explored the area to find suitable locations for more stations, expanding both northward and southward. In 1776, Jens Haven founded a second outpost at Okak, and in 1782, a third station named Hopedale was inaugurated. Detailed meteorological observations made by the Moravian

Brethren in Labrador are recorded in manuscripts kept in libraries and archives in both Europe and the New World (for details, see Demarée and Ogilvie, 2008). Unfortunately, few original measurements have survived. The history of meteorological measurements in Labrador made by Moravian missionaries in the late 18[th] century and later is presented in detail by Lüdecke (2005) and Demarée and Ogilvie (2008) and is thus omitted here.

According to our knowledge, our paper is the first modern work analysing thermal conditions in Labrador in the 18[th] century based on raw sub-daily meteorological data. We need to mention, however, that Demarée and Ogilvie (2008) present some temperature data for Nain in Labrador (only monthly maximum and minimum, see their Figure 4) taken from publications written by Titius and published in the *Wittenbergisches Wochenblatt* in the years from 1774 to 1786. Very limited

knowledge about the past climate in Labrador (including the study period) is available from temperature reconstructions using different kinds of proxy data (dendrochronological data, pollen analysis, oxygen isotopes, ice-rafted debris, etc.). The following latest papers can be given as examples: D'Arrigo et al., 2003; Viau and Gajewski, 2009; Richerol et al., 2014; Naulier et al., 2015; Alonso-Garcia et al., 2017; Rashid et al., 2023. In conclusion, this concise review summarises that the

state of knowledge regarding weather and climate in Labrador during the late 18th century is limited. That is why the main objective of this article is to describe temperature conditions in the coastal part of Labrador based on all available early instrumental data gathered by the Moravian missionaries. The eighteenth-century Moravian missionary observations offer a unique perspective on the climate of the Labrador coast, providing essential data on temperature (analysed in this paper), as well as about

atmospheric pressure, wind direction and force, and the general weather. By examining records in a detailed and systematic manner, as we did in this paper, a better understanding of the region's climate variability and trends during the early instrumental period will be available for comparisons with temperature reconstructions using proxy data and also with climate model simulations. In addition, it





will be possible to more reliably determine the range and direction of changes between historical and
contemporary periods than was also roughly done by Demaree and Ogilvie (2008).

We hope that the new knowledge presented here will also contribute to a better understanding
of the climate history in the whole of Canada and will also be useful for discussion about the
importance of historical climatology in the accurate identification of long-term climate changes.

**2 Area, data and methods**

**2.1 Area**

This paper focuses on analysing the thermal condition of NE Labrador's coastal region in the late 18[th]
century using instrumental meteorological observations made by Moravian missionaries at sites in
three settlements: Nain (1771–86), Okak (Okkak) (1776–87), and Hopedale (Hoffenthal) (1782–86)
(Fig. 1). Information about the precise geographical locations given in the sources (Fig. 2) is limited to
latitudes for Nain and Okak ( φ =56°30'N and  φ =57°30'N, respectively), while for Hopedale
approximate longitude is also given (φ=55°40'N, λ= c. 56°W). The measurements of longitude at that
time were not precise, and the prime meridian was also not agreed upon until 1884. That year, an
international conference (International Meridian Conference) was held in Washington, D.C. that
accepted the Greenwich Meridian as the universal prime meridian
(https://en.wikipedia.org/wiki/Prime_meridian_(Greenwich), access 1.08.2024). Thus, the given
values describing the geographical locations are not fully reliable and are not complete. Therefore, we
here give the locations of these stations (including altitudes) in the late 19[th] century after Hann (1896)
and Döll (1937): Nain, φ = 56°33'N, λ = 61°41'W, H = 4.2 m a.s.l.; Okak, φ = 57°34'N, λ = 62°03'W
(Hann), 61°56' (Döll), H = 7.5 m a.s.l.; Hopedale φ = 55°27'N, λ = 60°12'W, H = 7.6 m a.s.l.). The
geographical locations of the contemporary working meteorological stations at the airports in Nain and
Hopedale and of the location (central part) of the former Okak settlement (no meteorological station at
present) are the following: Nain (φ = 56°33' N, λ = 61°41'W, H = 6.4 m a.s.l.), Hopedale (φ =
55°27'N, λ = 60°13'W, H = 10 m a.s.l.), Okak (φ = 57°34' N, λ = 61°59'W, H = 3 m a.s.l.).



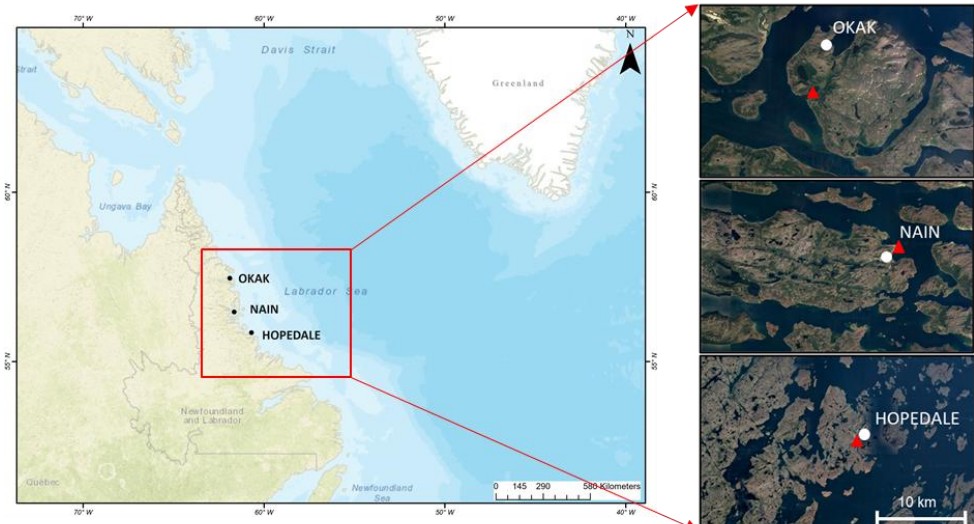


**Figure 1.** Study area and location of historical (circle) and contemporary (triangle) sites of meteorological measurements.

Source: Author's own work, based on https://www.google.pl/earth/. Map data for the location of sites: Okak , Imagery © 2024 CNES/Airbus,Maxar Technologies, Map data © 2024. Nain, Imagery © 2024 TerraMetrics, Map data © 2024 Google, Mapa GISrael. Hopedale, Imagery © 2024 Airbus, CNES/Airbus, Landsat /Copernicus,Maxar Technologies,Map data © 2024.

Of all three sites analysed in the paper, Okak is the most northerly on the coast of Labrador, on the largest Okak Island (north-west part of the archipelago of Okak Islands) at the mouth of the fjord (Fig. 1). Nain, the first mission established by the Moravian Church in 1771, is located on the north side of Unity Bay, a small inlet. The bay is open to the Atlantic Ocean, but Nain is protected by numerous islands (the largest of which is Paul's Island) and hilly terrain (Durkalec et al., 2014). From Nain to the open Labrador Sea is ~50 km. The historical station and the contemporary meteorological station situated in the airport (source of data for the period 1990–2020) are situated in close proximity to the coast, with the distance between them not exceeding 1 km. The southernmost site analysed in the paper is Hopedale. The settlement is situated on the tip of the peninsula, which is about 20 km long and surrounded by many islands, the largest being Annionwaktook Island. As in the case of Nain, the present meteorological station is located at the airport at less than 1 km from where the historical station was located (centre of the settlement) (see Fig. 1). The entire coast of this part of Labrador is of the skerry coast type, which strengthens maritime influences.

Climate conditions in this part of Labrador, and in the entire Labrador Peninsula, are poorly known before World War II because of the sparsity of data from the region (Barry, 1959). Therefore, any knowledge from before this time is crucial. The studied coastal part of the Labrador, according to Treshnikov (1985), belongs climatically to the Arctic. It is the southernmost part of the Arctic. The



climate here is very harsh, especially in winter, and significantly colder than in areas lying at the same
geographical latitude in Europe (e.g., southern Scandinavia) or the western coast of North America
(e.g., maritime Alaska from Juneau to Ketchikan). The climate in the coastal part of Labrador is so
cold mainly due to the influence of the cold Labrador Current. In winter, in addition, a great decrease
in air temperature is connected with atmospheric circulation (cold air masses incoming here within the

Canadian High-Pressure System). According to meteorological data that were at the disposal of
Treshnikov (1985), the analysed part of the coast of Labrador really had an Arctic climate, but now,
due to global warming, this area has a sub-arctic climate (Dfc, see https://koeppen-geiger.vu-
wien.ac.at/present.htm, last access 2.08.2024). For example, the mean temperature of the warmest
month (August) in the period 1991–2020 reached 11.1 °C (Nain) and 12.3 °C (Hopedale)

(*Environment and Climate Change Canada. Climate normals 1991-2020 station data*. Government of
Canada). On the other hand, as we wrote earlier, winters are very cold here. The mean temperature of
the coldest month (February) was -17.6 °C (Nain) and -16.4 °C (Hopedale). The absolute maximum
temperatures recorded in this time period exceeded 30 °C and reached 33.3 °C in July at both stations,
while the absolute minimum (February) was -40 °C and -41.4 °C in Hopedale and Nain, respectively.

Due to the relatively close proximity to the Icelandic Low, the values of precipitation are high here,
exceeding 800 mm, with a maximum in July and a minimum in May (*Environment and Climate
Change Canada. Climate normals 1991-2020 station data*. Government of Canada).

### 2.2. Data and methods

Thermal conditions of the coast of Labrador in the late 18[th] century were estimated based on
meteorological measurements made by the Moravian missionaries at the three sites. Examples of
sources of data are presented in Fig. 2. In light of what was stated in the previous sub-section, these
historical data, which are the oldest long-term instrumental observations (except Greenland, see
Przybylak et al. 2024) for anywhere in the entire Arctic, are crucial for improving knowledge of the

climate of Labrador prior to the beginning of regular observations in the region. The value of these
data for climatological studies is further increased by the fact that the series of air temperatures for
Nain (15 expedition years) and Okak (11 years) are more than 2 and 1.5 times longer, respectively,
than that for Nuuk in Greenland (7 years).





**Figure 2**. Examples of manuscripts presenting meteorological observations for (a) Nain (1 Oct 1771 to 31 July 1786), source: Unitatsarchiv MDF.1817, The Moravian Archives in Herrnhut (Germany), data presented in the manuscript: 1 to 14 January 1771, (b) Nain (22 Aug 1777 to 31 July 1786), source: R.S.MA 143, The Archives of the Royal Society in London, data presented in the manuscript: 28 Dec 1785 to 11 Jan 1786, (c) Okak (1 Aug 1779 to 31 July 1784), source: R.S.MA 144, The Archives of the Royal Society in London, data presented in the manuscript: 6 Sep 1780 to 17 Sep 1780, (d) Hopedale (1 Oct 1782 to 16 Aug 1786), source: R.S.MA 144, The Archives of the Royal Society in London, data presented in the manuscript: 25 March 1784 to 8 April 1784.

To study the thermal conditions of the coastal part of Labrador in the late 18th century, the sub-daily measurements have been used: Nain (Oct 1771 – Jul 1786), Okak (Okkak) (Oct 1776 – Aug 1787), and Hopedale (Hoffenthal) (Oct 1782 – Aug 1786). All these records were sourced from three primary archival collections: The Moravian Archives in Herrnhut (Germany), The Moravian Archives at Muswell Hill (United Kingdom), and the archives of the Royal Society in London (United Kingdom) (Fig. 2). Data coverage is shown in Fig. 3. The data are presented for "expedition years" (i.e., from September to August of the following year) because the meteorological registers were sent to Europe, mainly to Herrnhut in SE Germany (the main headquarters of missionaries), once a year in summer, when ships with different goods were able to reach the Moravian missionaries places in Labrador.





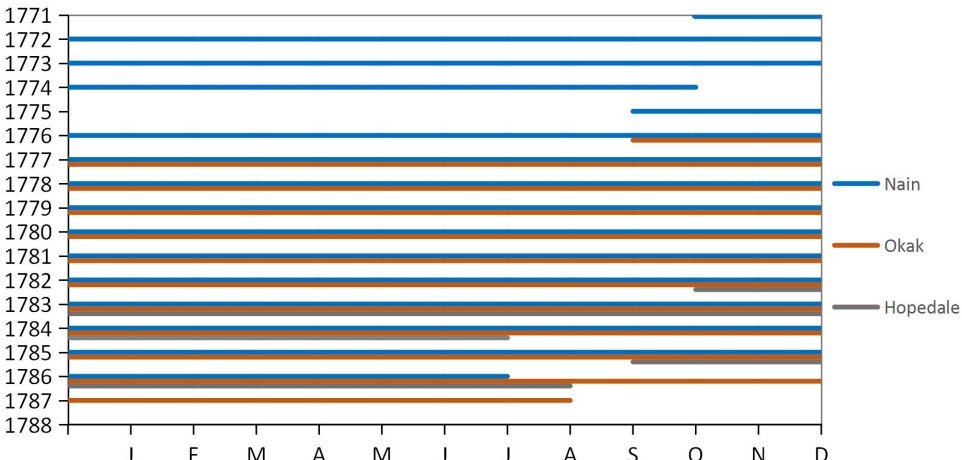

Fig. 3. Coverage of data available for Labrador for the late 18th century used in the present work.

To compare the historical data with contemporary ones, the latter were also gathered for all the study sites. For Nain and Hopedale, the data were taken from the meteorological stations working at the airports, while for Okak they were taken from the ERA5 for the nearest grid point (Table 1, Fig. 1). The quality of ERA5 data for this place was checked by comparison of ERA5 data against instrumental data available for Nain station. Differences in mean temperatures calculated based on

these two sets of data from the period 1991–2012 were used to correct the ERA5 data for Okak. The existing gaps in daily data for Hopedale were filled using data from Nain by comparison of a common series of 11 years of data (constant difference method). The Pearson correlation coefficient of daily data between these two stations is very high and is equal to 0.987.

**Table 1.** Sources of meteorological data for Nain, Okak and Hopedale, Sep 1990 – Aug 2020.

| Station | Source | |
|---|---|---|
| Nain | Canadian Centre for Climate Services, https://climate.weather.gc.ca/historical_data/search_historic_data_e.html, gaps in observations filled using data from Hopedale or corrected values from ERA5 | |
| Okak (grid: φ =57°50'N λ = 62°00'W | ERA5 reanalysis data: https://cds.climate.copernicus.eu/cdsapp#!/dataset/reanalysis-era5-complete?tab=overview, corrected based on comparison of instrumental data in Nain and ERA5 | |
| Hopedale | Canadian Centre for Climate Services, https://climate.weather.gc.ca/historical_data/search_historic_data_e.html, gaps in observations filled using data from Nain or corrected values from ERA5 | |



Prior to conducting statistical analyses, all available historical data sets underwent thorough quality checks and necessary corrections. For instance, air temperature measurements taken with a Fahrenheit thermometer during the period of 1771–86 (Nain), 1776–87 (Okak) and 1782–86 (Hopedale) were converted to degrees Celsius to ensure consistency with current units. During this period, measurements were primarily recorded in Nain at 08:00 and 14:00, but there were days with three, four or five measurements daily. In Okak and Hopedale, a standardised measurement schedule was implemented, with readings consistently taken at 08:00, 12:00, 16:00 and 20:00. In the next step, mean daily air temperature (MDAT) values for the historical periods were computed using nine different formulas listed below ([2]–[10]). Using contemporary data from Nain and Hopedale from the period 1991–2000, we also calculated MDAT for all the above formulas and for formula (1). This formula uses 24 hourly observations per day, allowing us to calculate the "real" MDAT. To identify biases in calculating MDATs using formulas (2)–(10), the differences between them and MDAT1 (formula [1]) were calculated separately for each month (Table 2). In this way, calculated corrections were added to historical MDAT2–MDAT10 values to obtain corrected values of MDAT for this period. In Okak (no instrumental data available at present), the corrections were taken from Nain (Table 2). By taking these precautions, the temperature data used in the statistical analysis were guaranteed to be accurate and reliable.

Below are the formulas that we used to calculate the historical MDAT for the studied stations:

$$\text{MDAT1} = (T1+T2+T3+ \ldots T24)/24 \tag{1}$$

$$\text{MDAT2} = (T8+T14)/2 \tag{2}$$

$$\text{MDAT3} = (T8+T14+T21)/3 \tag{3}$$

$$\text{MDAT4} = (T3+T8+T14+T16)/4 \tag{4}$$

$$\text{MDAT5} = (T4+T8+T12+T14+T16)/5 \tag{5}$$

$$\text{MDAT6} = (T6+T8+T12+T14+T16)/5 \tag{6}$$

$$\text{MDAT7} = (T6+T8+T12+T16+T20)/5 \tag{7}$$

$$\text{MDAT8} = (T7+T12+T16+T20)/4 \tag{8}$$

$$\text{MDAT9} = (T8+T12+T16+T20)/4 \tag{9}$$

$$\text{MDAT10} = (T8+T12+T14+T16)/4 \tag{10}$$



**Table 2.** Air temperature corrections (°C) used to calculate "real" MDAT for Nain, Okak and Hopedale for historical periods, 1771–87.

| Formula | J | F | M | A | M | J | J | A | S | O | N | D |
|---|---|---|---|---|---|---|---|---|---|---|---|---|
| Nain and Okak | | | | | | | | | | | | |
| MDAT2-MDAT1 | 0.26 | 0.33 | 0.59 | 0.80 | 1.05 | 1.37 | 1.28 | 1.25 | 1.08 | 0.63 | 0.30 | 0.25 |
| MDAT3-MDAT1 | 0.11 | 0.23 | 0.42 | 0.52 | 0.48 | 0.59 | 0.58 | 0.51 | 1.46 | -0.09 | 0.09 | 0.11 |
| MDAT4-MDAT1 | 0.32 | 0.58 | 0.74 | 0.77 | 0.84 | 0.97 | 0.92 | 1.03 | 0.96 | 0.60 | 0.31 | 0.25 |
| MDAT5-MDAT1 | 0.30 | 0.55 | 0.69 | 0.71 | 0.80 | 1.01 | 0.92 | 1.00 | 0.91 | 0.56 | 0.31 | 0.25 |
| MDAT6-MDAT1 | 0.29 | 0.50 | 0.63 | 0.73 | 0.97 | 1.23 | 1.11 | 1.12 | 0.88 | 0.55 | 0.29 | 0.23 |
| MDAT7-MDAT1 | 0.02 | 0.09 | 0.19 | 0.27 | 0.46 | 0.61 | 0.56 | 0.47 | 0.30 | -0.20 | 0.04 | 0.04 |
| MDAT8-MDAT1 | 0.19 | 0.49 | 0.65 | 0.75 | 0.78 | 0.91 | 0.90 | 0.82 | 0.69 | -0.03 | 0.19 | 0.17 |
| MDAT9-MDAT1 | 0.19 | 0.52 | 0.79 | 0.94 | 0.94 | 1.10 | 1.06 | 1.02 | 0.88 | 0.10 | 0.20 | 0.17 |
| MDAT10-MDAT1 | | | | | | | | | 1.62 | 0.97 | | |
| Hopedale | | | | | | | | | | | | |
| MDAT9-MDAT1 | 0.26 | 0.50 | 0.66 | 0.84 | 0.66 | 0.64 | 0.57 | 0.59 | 0.60 | 0.32 | 0.16 | 0.18 |

Both the original and corrected MDAT values for all stations are available at NCU https://doi.org/10.18150/VJJFOE, Singh et al.,2024). In addition to standard climate metrics (monthly, seasonal, and annual means, etc.), several specialised climate indices were computed using the corrected MDAT values. These included growing degree days (GDD), air thawing index (ATI), positive degree days (PDD), and air-freezing index-degree days (AFI) (Table 3). The methodologies for calculating the latter four indices were based on the definitions provided by Nordli et al. (2020),

while their importance in studies of climate and environment was briefly summarised in our previous paper (Przybylak et al., 2024).

**Table 3.** Definitions of terms used in threshold statistics (after Nordli et al. 2020).

| Terms | Definitions |
|---|---|
| Annual growing degree-days sum | $GDD = \sum_{i=1}^{n} Max(0, T_i - 5) \, for \, May - Sep$ |
| Air thawing index degree-days sum | $ATI = \sum_{i=1}^{n} Max(0, T_i) \, for \, May - Sep$ |
| Positive degree-days sum | $PDD = \sum_{i=1}^{n} Max(0, T_i) \, for \, Oct - Apr$ |
| Air freezing index degree-days sum | $AFI = \sum_{i=1}^{n} Min(0, T_i) \, For \, Oct - Apr$ |
| $T_i$ | Mean temperature on day $i$ and $n$ is the number of days |
| $n$ | Number of days |

The frequency of occurrence of MDAT was calculated for four standard seasons (DJF, MAM, etc.) for

one-degree intervals. To assess whether the distributions of MDAT conform to a normal distribution, their skewness ($\gamma 1$) and kurtosis ($\gamma 2$) were calculated. These statistical measures were computed using the methodologies proposed by von Storch and Zwiers (1999).



An equally important statistic is the frequency of MDATs exceeding chosen thresholds, which
allow us to distinguish "characteristic" days (see the criteria used for this below). Usually, for this
purpose, extreme daily values of air temperature (Tmax and Tmin) are utilised. However, extreme
thermometers were constructed only at the end of the 18th century, in 1782 (Tamulewicz, 1997, p. 45)
and, therefore, could not be used by the Moravian missionaries in Labrador.

The criteria that have been used to distinguish characteristic days are the following:

1. MDAT >20 °C    – exceptionally warm day
2. MDAT >15 °C    – extremely warm day
3. MDAT >10 °C    – very warm day
4. MDAT >5 °C      – warm day
5. MDAT ≥0 °C      – non-frost day
6. MDAT <0 °C      – frost day
7. MDAT <-5 °C    – slightly cold day
8. MDAT <-10 °C   – moderately cold day
9. MDAT <-15 °C   – cold day
10. MDAT <-20 °C – very cold day
11. MDAT <-25 °C – extremely cold day
12. MDAT <-30 °C – very extremely cold day
13. MDAT <-35 °C – exceptionally cold day

To estimate the continentality of the climate on the coast of Labrador, an index proposed by
Ewert (1972) has been used. The continentality index (K) proposed by Ewert was calculated according
to the formula:

$$K = [ATR - (3.81 \cdot \sin \varphi + 0.1) / (38.39 \cdot \sin \varphi + 7.47)] \, 100\%$$

where ATR is the annual temperature range, and $\varphi$ is the geographical latitude.

## 3 Results

### 3.1 Monthly resolution

The annual courses of historical air temperatures from 1771–86 (Nain), 1776–87 (Okak), and 1782–86
(Hopedale) have been compared to the 30-year means of the contemporary period Sep 1990 – Aug
2020 (Table 4, Fig. 4). In both historical and contemporary periods, the warmest month in the coastal
part of the Labrador was August and the coldest was February (except Hopedale in the historical
period). This suggests a significant influence of the Atlantic Ocean on the climate of the coast of
280   Labrador. The annual range of temperature (warmest minus coldest monthly temperature) in Nain
reached 29.4 °C in the historical period and 28.7 °C in the present. This means that climate



continentality in the late 18<sup>th</sup> century was slightly higher than at present (Ewert's index of climate continentality was equal to 66.7% and 64.9%, respectively).

The historical period (1771–86) in Nain was, on average, 2.3 °C colder than today. The cooling was greatest in winter (DJF), with an anomaly reaching -3.5 °C, and smallest in spring (MAM), at only -0.8 °C (Table 4, Fig. 4). A generally similar pattern in the change of the annual cycle of temperature between the historical and present periods was also observed in Okak and Hopedale. However, we must note that, at these places, the mean annual air temperature was only 0.5–0.7 °C colder than today and that the spring temperature was even greater than at present (by 1.3 °C and 2.3 °C for Okak and Hopedale, respectively). All these changes are probably related to the differences in analysed historical periods in comparison to Nain. Generally, the historical mean monthly data lie within one standard deviation of present means, except for autumn values in Nain, when the temperature exceeds slightly that threshold (Fig. 4). This indicates that historical temperatures lie completely within the range of their modern variability.

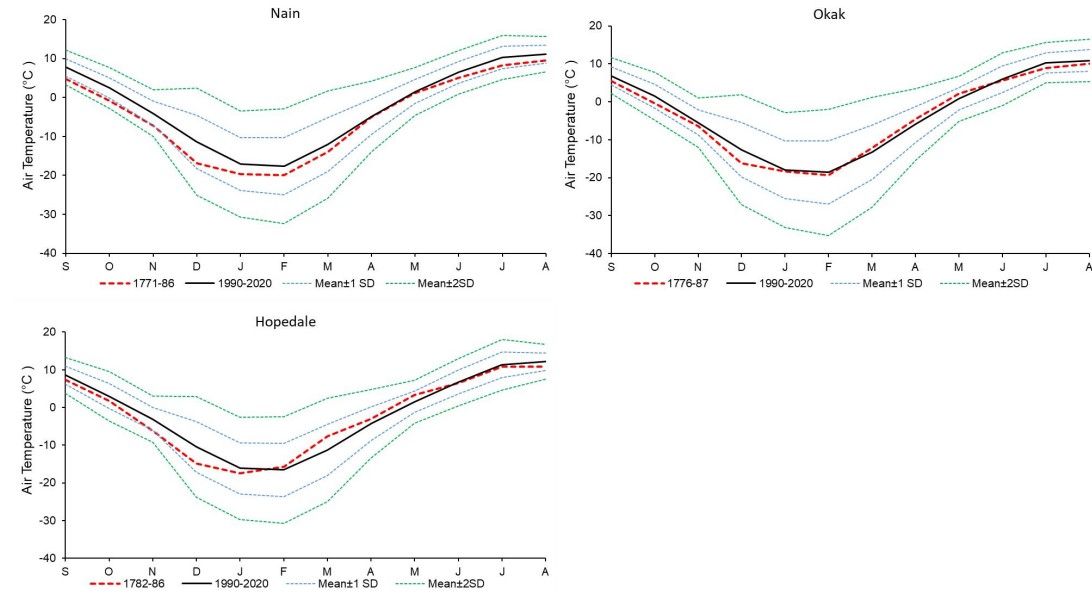

**Figure 4.** Annual courses of historical and modern air temperatures in Nain, Okak and Hopedale based on monthly means. SD calculated based on present data (Sep 1990 – Aug 2020)



**Table 4.** Mean monthly, seasonal and annual air temperature in Nain, Okak and Hopedale in historical (H) and the contemporary (C) periods and their differences

| Station | Year | S | O | N | D | J | F | M | A | M | J | J | A | SON | DJF | MAM | JJA | Year |
|---|---|---|---|---|---|---|---|---|---|---|---|---|---|---|---|---|---|---|
| Nain | H 1771–86 | 4.7 | -0.8 | -7.2 | -16.9 | -19.7 | -19.9 | -14.0 | -5.1 | 1.1 | 5.0 | 8.3 | 9.5 | -1.1 | -18.8 | -6.0 | 7.6 | -4.6 |
| | C 1990–2020 | 7.8 | 2.5 | -4.0 | -11.4 | -17.1 | -17.6 | -12.1 | -4.9 | 1.6 | 6.5 | 10.3 | 11.1 | 2.1 | -15.4 | -5.1 | 9.3 | -2.3 |
| Okak | H 1776–87 | 5.4 | -0.5 | -6.5 | -16.2 | -18.4 | -19.3 | -12.2 | -4.6 | 2.1 | 5.6 | 8.9 | 10.1 | -0.5 | -18.0 | -4.9 | 8.2 | -3.8 |
| | C 1990–2020 | 6.9 | 1.5 | -5.4 | -12.6 | -17.9 | -18.6 | -13.3 | -6.0 | 0.8 | 6.0 | 10.3 | 10.9 | 1.0 | -16.4 | -6.2 | 9.1 | -3.1 |
| Hopedale | H 1782–86 | 7.3 | 1.7 | -6.1 | -14.9 | -17.5 | -15.8 | -7.6 | -3.0 | 3.4 | 6.6 | 10.8 | 10.9 | 0.9 | -16.1 | -2.4 | 9.4 | -2.0 |
| | C 1990–2020 | 8.6 | 3.0 | -3.0 | -10.5 | -16.1 | -16.6 | -11.3 | -4.3 | 1.5 | 6.7 | 11.4 | 12.2 | 2.8 | -14.4 | -4.7 | 10.1 | -1.5 |
| | | | | | | | | Differences* | | | | | | | | | | |
| Nain | H 1771–86 C 1990–2020 | -3.0 | -3.3 | -3.2 | -5.5 | -2.6 | -2.3 | -1.9 | -0.1 | -0.5 | -1.5 | -2.0 | -1.7 | -3.2 | -3.5 | -0.8 | -1.7 | -2.3 |
| Okak | H 1776–87 C 1990–2020 | -1.5 | -1.9 | -1.0 | -3.6 | -0.5 | -0.7 | 1.1 | 1.4 | 1.3 | -0.4 | -1.4 | -0.8 | -1.5 | -1.6 | 1.3 | -0.9 | -0.7 |
| Hopedale | H 1782–86 C 1990–2020 | -1.2 | -1.3 | -3.1 | -4.5 | -1.4 | 0.8 | 3.7 | 1.3 | 1.9 | -0.2 | -0.6 | -1.3 | -1.9 | -1.7 | 2.3 | -0.7 | -0.5 |

*Values from period C were subtracted from period H.

Out of all 15 expedition years available for Nain, the warmest year was 1779/80 (-1.7 °C), and the coldest was 1773/74 (-8.5 °C). In the same sets of years taken from the contemporary period (Sep 1990 – Aug 2005 and Sep 2005 – Aug 2020), the warmest mean annual temperatures reached -0.6 °C (2003–04) and 0.7 °C (2009–10), respectively, while the coldest -4.9 °C (1990–91) and -3.5 °C (2014–15), respectively. It means that both the warmest and coldest individual years are warmer at present

than in the historical period.

     The year-to-year variability of mean monthly air temperatures in Nain (1771–86), as estimated using standard deviation (SD), in the historical period was greatest in winter months, particularly in January (5.7 °C) and smallest in summer months (1.5 °C in June and 1.6 °C in August) (Fig. 5). A similar variability in the annual cycle occurred also in Okak (1776–87), with the highest value in

January (6.2 °C) and the lowest in June (0.7 °C). Fig. 5 shows that mean monthly temperatures are more stable from year-to-year in Okak than in Nain. In both sites, particularly in Nain, variability was greater in the historical period than at present (1991–2020). Only in August and June were the values similar to or lower than the contemporary variability. In the case of Okak, such a situation occurred also from May to August.



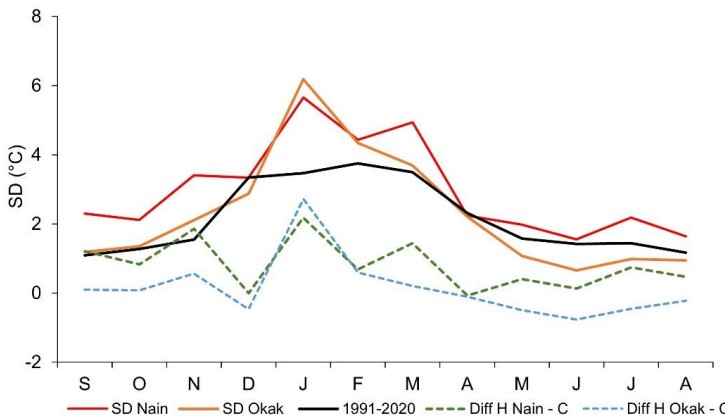


**Figure 5.** Annual cycle of SD based on monthly data for Nain and Okak for historical (H, colour lines) and contemporary (C, black line) periods and their differences (Diff).

To widen the analysis of temperature change between historical and present periods, we also compared temperature indices (GDD, ATI, PDD, and AFI) calculated using the MDAT data (Figs 6, 7

and 8). In Nain and Okak (Figs 6 and 7), the historical mean GDD levels are lower than the recent averages but higher than their minimum values. This means that, today, we have a longer and warmer vegetation period than in the late 18th century. A more complicated picture presents the Hopedale site (Fig. 8), where GDD in May, June and July was the same in both periods (historical and present), while in August and September the GDD was clearly higher at present than in the historical period.

For all studied sites, the ATI values show similar behaviour as the GDD. Both in the historical and the modern periods the PDD values are near zero from November to March, slightly higher in early spring (April), and largest in October (Figs 6, 7 and 8). In all months, the PDD is at present equal to or greater than values calculated for the historical periods, except March and, in particular, April for Okak and Hopedale. Compared to the recent decades, historical AFI levels are often higher, indicating

more intensive and frequent freezing weather in Nain. On the other hand, in Okak and Hopedale, such a pattern occurred only from October to February, while in March and April, the AFI was lower in the historical periods in comparison to the present.



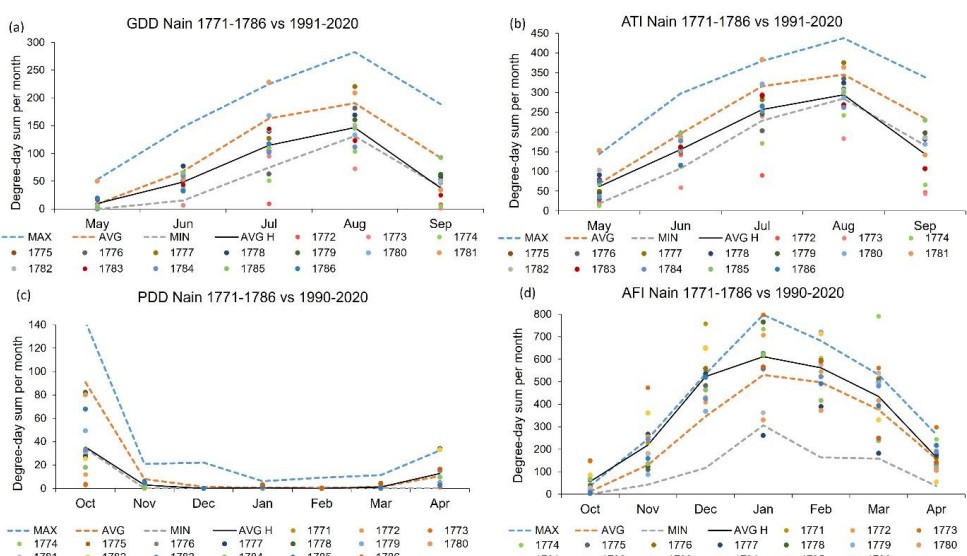

**Figure 6.** Comparison of temperature indices (GDD, ATI, PDD and AFI) calculated for Nain for historical
(1771–86, black lines) and contemporary (Sep 1990 – Aug 2020, colour lines) periods.

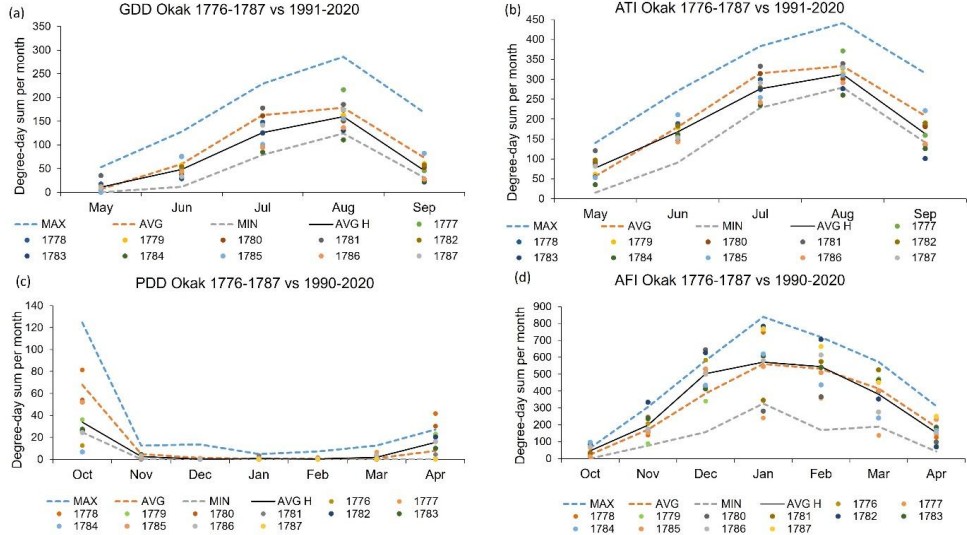

**Figure 7.** Comparison of temperature indices (GDD, ATI, PDD and AFI) calculated for Okak for historical
(1776–87, black lines) and contemporary (Sep 1990 – Aug 2020, colour lines) periods.



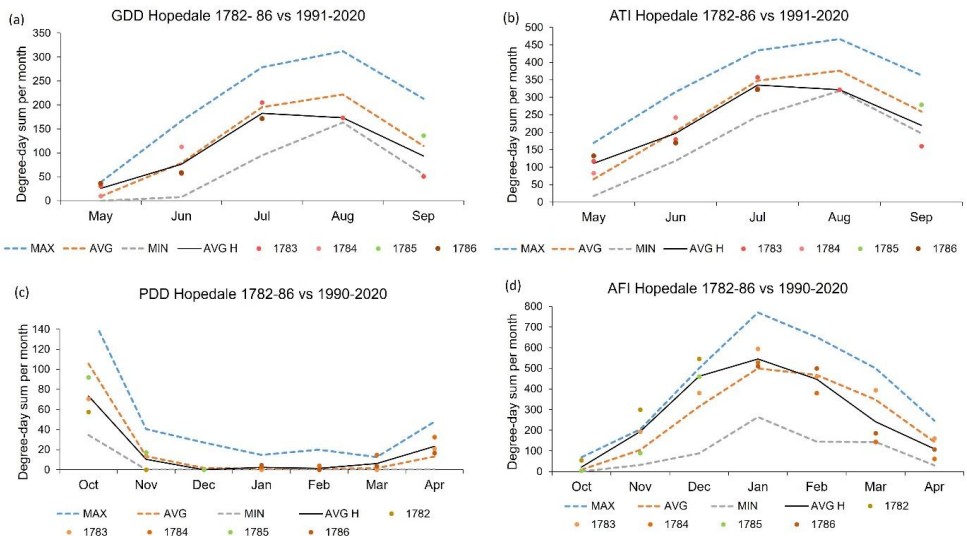

**Figure 8.** Comparison of temperature indices (GDD, ATI, PDD and AFI) calculated for Hopedale for historical (1782–86, black lines) and contemporary (Sep 1990 – Aug 2020, colour lines) periods.

### 3.2 Daily resolution

In our previous studies for different Arctic regions (Canadian Arctic, Novaya Zemlya, Svalbard, Greenland), we showed the advantages of using daily resolution air temperature in regional climate analysis as compared to monthly resolution (Przybylak and Vizi 2005; Przybylak and Wyszyński 2017; Przybylak et al. 2018, 2024; Nordli et al. 2020). The same methods and climate indices used in these works allow for reliable comparison of results and, therefore, also to identify time and spatial changes in different climate characteristics. That is why we decided also in the analysis of Labrador climate to use this kind of data. The analysis of the annual courses of historical MDATs for the Labrador coast reveals that they were both greater and smaller than the values of MDATs in the contemporary period (1990–2020). It is worth noting, however, that the MDAT values for the longest and, therefore, most reliable historical period, 1771–86 (Nain), were very rarely higher than at present. In the other two analysed historical periods (1767–87 and 1782–84), warmer spells in MDAT data were more frequent than in the first-analysed period and were particularly common in spring (Fig. 9). The greatest fit between MDATs in historical and present periods exists for spring in Nain (1771–86) and in summer for the two other sites. On the other hand, the greatest discrepancy between curves is noted in winter months, particularly in December and January (Fig. 9). Notably, most MDAT data from Nain, Okak and Hopedale do not exceed 1 SD from the present-day means of MDATs and very rarely exceed 2 SD (Fig. 9).





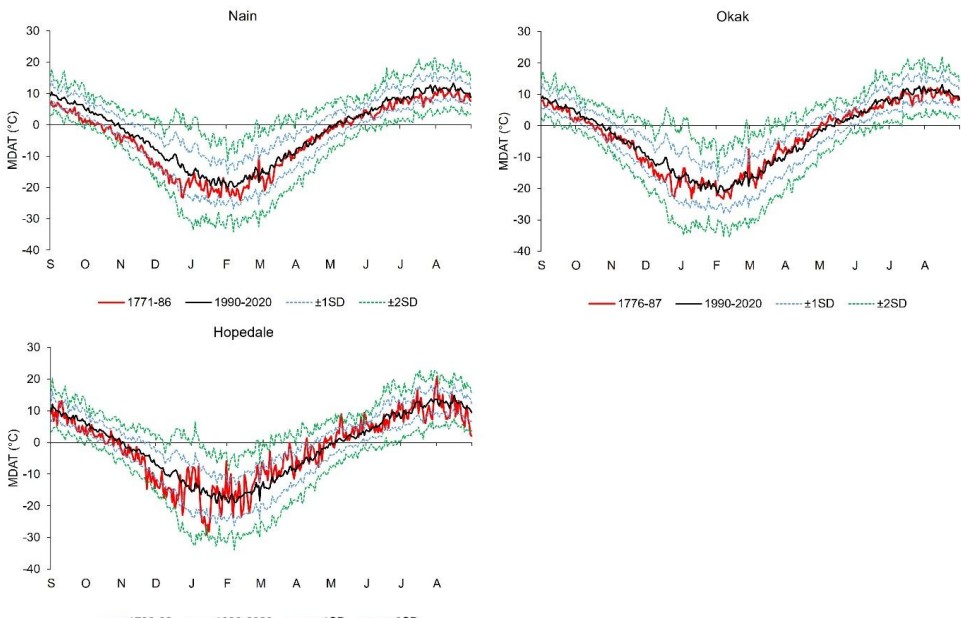

**Fig. 9**. Annual courses of MDAT on the coast of Labrador in historical years (lines in different colours) and in
the contemporary period (Sep 1990 – Aug 2020) (black line). Blue (±1 SD) and green (±2 SD) dashed lines
indicate SD calculated for 1990–2020 and added to or subtracted from the present mean.

Averaging can sometimes significantly obscure some climatic details. To address this, we provide a
more detailed representation of the climate conditions in Labrador in the late 18th century utilizing
various climate indices and characteristics based on daily data. One is the relative frequencies of
occurrence (in %) of MDAT in the historical and contemporary periods. The relative frequencies of
occurrence of air temperatures were stratified into one-degree intervals (Fig. 10). The MDATs for the
four seasons in Nain typically follow a near-normal distribution with skewness ($\gamma1$) values falling
between -0.7 and 0.4, and kurtosis ($\gamma2$) between about -0.6 and 1.0. Of all seasons, the MDAT
distribution is closest to normal in summer, followed by winter, in historical and contemporary periods
alike. The greater frequency of warmer temperatures at present in comparison to the historical period
is observable mainly in autumn and then in summer. On the other hand, the greatest fit between curves
showing MDAT for present and historical periods occurred in spring (Fig. 10). It is also worth noting
the platykurtic distribution of MDAT data in the spring and winter. Generally, the frequency of
MDAT occurrences in Okak (Fig. S1) in all analysed seasons reveals a very similar distribution to that
presented for Nain (compare Figs 10 and S1). This statement is also confirmed by slightly different
values of skewness and kurtosis calculated using MDATs for both sites. Figure S2 shows the relative
frequency of occurrence of MDAT in Hopedale. For this place, we have the shortest and least
representative series for the late 18[th] century, and this therefore probably accounts for some differences
noted in comparison to the earlier-described MDAT distributions for Nain and Okak. Small changes





are noted in the winter half-year, and great changes in the summer half-year (especially spring). In this season in Hopedale in the historical time (1782–86), warmer temperature classes than today were observed. On the other hand, in summer, a bi-modal distribution is observable in the historical period (Fig. S2). But despite these differences in comparison to Nain and Okak temperature distributions, the MDAT distributions in Hopedale are close to normal in all seasons. The skewness values oscillate between -0.6 and 0.5 and kurtosis between -0.7 and 0.4. Fig. S2 and, in particular, values of kurtosis document that the platykurtic distribution dominates in the contemporary period in all seasons (though particularly in winter). In winter, that distribution is also clearly evident in the historical period.

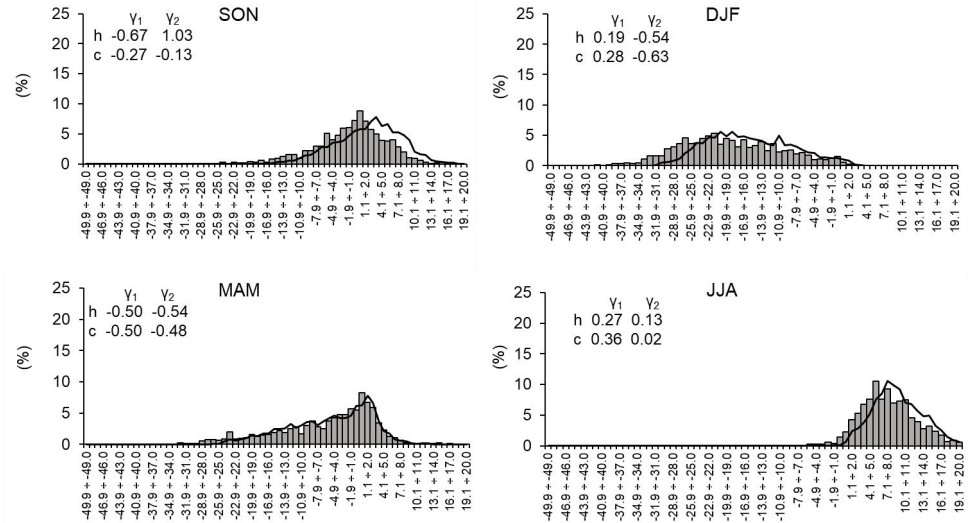

**Figure 10.** Seasonal (Sep–Nov, Dec–Feb, etc.) relative frequencies of occurrence (in %) of MDAT in Nain in historical (1771-86, bars) and modern (Sep 1990 – Aug 2020, lines) periods. Values of skewness ($\gamma 1$) and kurtosis ($\gamma 2$) for historical (h) and contemporary (c) times are also shown.

The relative frequency of occurrence of different characteristic days in Nain in the study historical period is shown in Figure 11. The extremely cold days (MDAT <-25 °C), very extremely cold days (MDAT <-30 °C), and exceptionally cold days (MDAT <-35 °C) occurred mainly from December to March, and sporadically in other months of the cold half-year. The rest of the categories of cold days (warmer than previous ones) shown in Fig. 11 were most common in a longer period, i.e. in the entire cold half-year (from November to April). All categories of warm days (i.e., warm, very warm, extremely warm and exceptionally warm) were noted in Nain in the historical period, mainly from June to September, with a peak in August. In May and September, some categories of warm days were also observed but with small frequency. From May to September, positive MDATs dominated, while for the rest of the year – negative ones (Fig. 11). However, it is worth noting that both positive and



negative MDAT values occurred in all months. A comparison of Fig. 11 with Figs S3 and S4,
presenting the frequency of occurrence of characteristic days in Okak and Hopedale, respectively,
indicates that all distributions are very similar.

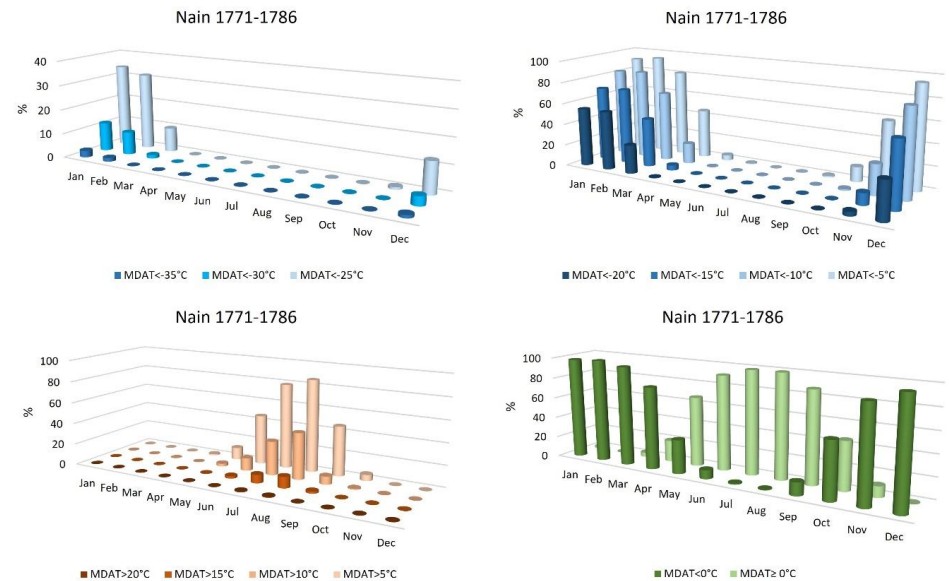

**Figure 11.** Annual courses of relative frequency of occurrence (in %) of characteristic days in Nain in historical
time.

In line with expectations, the overall number of all categories of cold days decreased from the
historical period to the modern period, particularly in December (Figs 12–14). The opposite tendency
for moderately cold days was noted in Okak (Fig. 13) in the period from January to April and in
Hopedale (Fig. 14) for February and March.

Thus, these statistics indicate that more harsh winter conditions were observed in historical
times than today. In contrast, the modern period is characterised by a significant increase in all
categories of warm days, particularly during the summer months, but also in September and less so in
October. Only in May (Okak) and in April and May (Hopedale) warm (MDAT>5 °C) and very warm
(MDAT>10 °C) days were more frequent in historical time than at present (Figs 12–14).





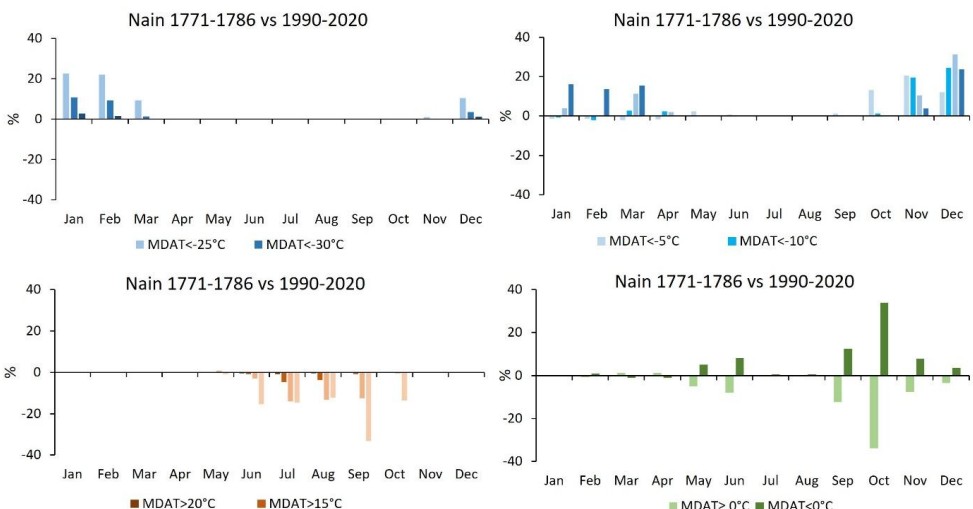

**Figure 12**. Annual courses of differences between the number of characteristic days (in %) in Nain in historical and the contemporary (Sep 1990 – Aug 2020) periods. Differences were obtained by subtracting contemporary values from historical ones.

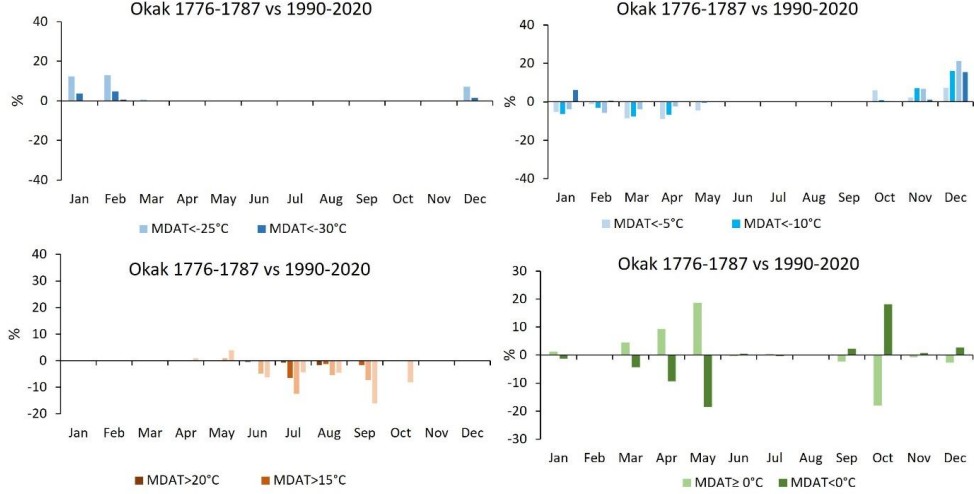

**Figure 13**. Annual courses of differences between the number of characteristic days (in %) in Okak in historical and the contemporary (Sep 1990 – Aug 2020) periods. Differences were obtained by subtracting contemporary values from historical ones.



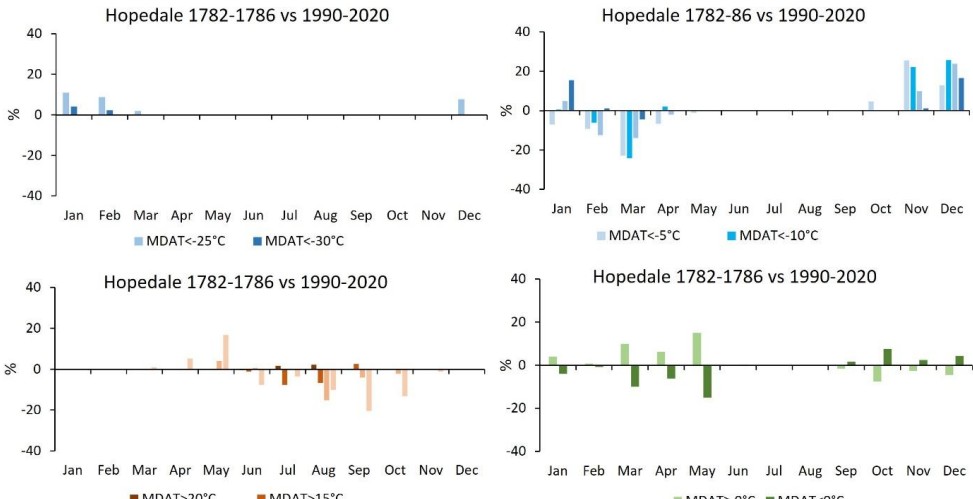

**Figure 14.** Annual courses of differences between the number of characteristic days (in %) in Hopedale in
historical and the contemporary (Sep 1990 – Aug 2020) periods. Differences were obtained by subtracting
contemporary values from historical ones.

## 4 Discussion

In the late 18th century, the climate in the Arctic, including Labrador, was governed mainly by natural
factors, i.e. income of solar radiation and magnitude of volcanic activity (e.g., Bond et al., 2001; Miller
et al. 2012), as well as by inner variability of the climatic system (Wanner et al., 2011). The study
period (1771–87) is part of the Little Ice Age (LIA). The existing reconstructions of ocean and land
temperatures show that the LIA cooling was neither spatially nor temporally uniform (e.g., Overpeck
et al., 1997; Bradley et al., 2003; PAGES 2k Consortium, 2013; Wanner et al., 2011, 2015). For
example, our previous studies for SW Greenland (Przybylak et al., 2024), also based on instrumental
data measured by Moravian missionaries, have shown that the late 18th century (period 1784–92)
experienced significant warming, comparable to or even warmer than the Early Twentieth-Century
Arctic Warming period but colder than today. Similar results were presented also by Hörhold et al.
(2023) based on a reconstruction of central and northern Greenland temperatures stacked from a
compilation of 21 stable oxygen isotope $\delta^{18}O$ records. They showed (see their Fig. 1a) the existence of
a warm period in the second half of the 18th century of comparable magnitude to that occurring during
the MWP but colder than the recently observed temperature. But there are also some studies (e.g.,
Kobashi et al., 2010) showing that in central Greenland the temperature in the second half of the 18th
century was the coldest in the entire millennium. Labrador, including its coast, is also not free from
similar inconsistencies in the air temperature (or sea surface temperature [SST] in the Labrador Sea)
reconstruction results obtained for the period under study. For example, there exists only one paper
(Demarée and Ogilvie, 2008) giving some information about the temperature on the Labrador coast in



the 18th century (see Introduction). The authors did not show a quantitative comparison with the present thermal conditions. Nevertheless, they concluded that: "A preliminary comparison with
present-day climatic observations suggested no systematic differences between the early instrumental observations of the eighteenth and nineteenth centuries and those of the twentieth." On the other hand, quite a lot of reconstructions are available based on proxy data. Examples of some of the most important of them follow. Viau and Gajewski (2009) reconstructed January and July temperatures from the fossil pollen records using the modern analogue technique. July temperature was one of the
lowest temperatures in the entire last millennium, while January was colder than in medieval times but warmer than at present (see their Fig. 2). Tree-ring reconstructions of the summer temperature are not consistent also, see sites 25 and 26 in Fig. 2 available in Overpeck et al. (1997). In the last three decades of the 18th century, there were great cold summers in Okak (site 25), while in a nearby location (Salt Water Pond, site 26), summers were slightly warmer. A clearly warm Jun–Sep period
was also found by D'Arrigo et al. (2003), who used both tree-ring widths and maximum latewood density data from many sites lying in the coastal part of Labrador (from Eyeglass and Okak, in the north to Salt Water Pond in the south, see their Fig. 1). Also, the warm period in the second half of the 18th century in the Labrador Sea was estimated by analysis of ice-rafted debris (Alonso-Garcia et al., 2017). A multi-proxy approach (fossil dinoflagellate cysts, diatoms and pollen) utilized by Richerol et
al. (2014) for sediment cores taken from three Labrador fjords (Nachvak 59°N, Saglek 58.5°N, and Anaktalak 56.5°N) indicates similar results. SST reveals strong stability in the last 400 years, but its values in the second half of the 18th century were similar to the present. The reconstructed sea-ice cover confirms this finding, the sea-ice cover in both periods was comparable (see their Fig. 10). The most recent reconstruction of SST from an area lying close to the Labrador coast (Placentia Bay, NE
Newfoundland Shelf) also confirm the existence of warm conditions at this time (Rashid et al., 2023). It is worth adding that, according to most recent reconstructions of mean summer or annual air temperatures in the Arctic (Overpeck et al., 1997; Kaufman et al., 2009; Hanhijärvi et al., 2013; McKay and Kaufman, 2014; Werner et al., 2018), the second half of the 18th century was warmer than most of the years belonging to the LIA (1600 to 1900), as it was in the entire Northern Hemisphere
(e.g., Moberg et al., 2005; Hegerl et al., 2007). From the presented review results, there is very little work analysing instrumental data for this period. Significantly more abundant are reconstructions of air temperature, SST or other variables related to climate (mainly temperature). The majority of reconstructions described above revealed the existence of warming in the second half of the 18th century. The question is whether the results presented here for Labrador based on instrumental
observations are in agreement with that conclusion. The answer is not easy and not clear because reconstructions using proxy data present the results for different sub-periods of the warm half-year, and the reconstructions are "generally limited to multi-year temporal resolution, annual in the best cases" (Ouellet-Bernier et al., 2021). On the other hand, the available series of sub-daily temperature observations made by Moravian missionaries, although relatively long, by instrumental standards, are



short in comparison to reconstructions based on proxy data. Moreover, as temperature reconstructions based on proxy data show, the second part of the 18th century was not stable. For example, D'Arrigo et al. (2003) found that the period Jun–Sep was colder in the 1770s than in the 1780s (reconstruction based on tree-ring widths). Instrumental data from Nain, particularly from the beginning of the first decade (1771–74), fully confirm this statement. According to reconstruction based on wood density, a

warm wave occurred from about 1775 until 1815 (see Fig. 2 in D'Arrigo et al. 2003). This is one of the reasons why the temperature from Nain (1771–86) was colder than in two other sites (Okak and Hopedale), having more data from the mentioned warm wave (see Table 4). The calculation for a common ten-year period (Sep 1776 – Aug 1786) reveals the same mean annual temperature (-3.7 °C) in Nain and Okak. Summer and, in particular, autumn were slightly warmer in Nain, while winter and

spring were warmer in Okak. Therefore for comparison of historical temperature with more modern data, only data from Nain are shown (Fig. 15). In addition,

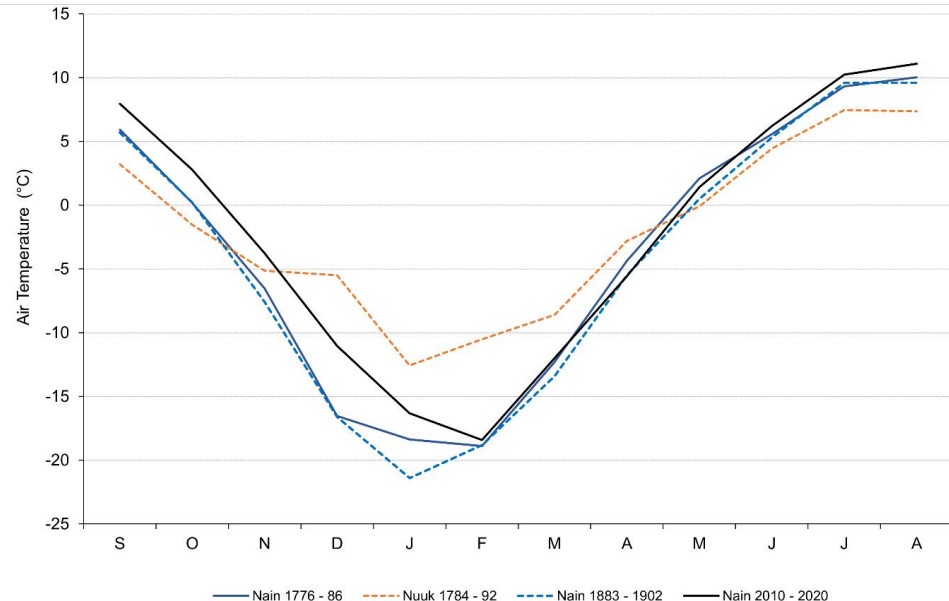

**Fig. 15**. The annual cycle of air temperature in the Labrador coast (Nain) and SW Greenland (Nuuk) for historical and contemporary periods (presented for expedition years). Data from Nuuk are taken from Przybylak

et al. (2024), and data from Nain for the period 1883–1902 after Döll (1937).

data from Nuuk (SW Greenland) from the historical period (1784–92) are also presented. It is very well seen that the climate is more continental on the Labrador coast than on the coast of SW Greenland (Fig. 15). The mean winter temperature in Nain was 8.6 °C colder than in Nuuk, while the summer was

1.9 °C warmer. The mean annual temperature was 1.7 °C higher in Nain than in Nuuk. We also compared the temperature in Nain in the study period (1776–86) with the temperature observed in periods 1883–1902 and 2010–20 (Fig. 15). The courses of temperature curves in the first two periods



are very similar, and usually, the differences are below or slightly above 1 °C (except January, 3 °C). On average, Nain was 0.7 °C warmer in the late 18th century than at the turn of the 19th century, with the greatest difference in autumn (1.3 °C) and the smallest in summer (only 0.1 °C). On the other hand, in comparison to the contemporary period (2010–20), the temperature in Nain was, on average, 1.4 °C lower in the study historical period. The greatest rise in temperature from the historical to the contemporary period occurred in the cold half-year (~2.5 °C). In summer, it reached only 0.9 °C, while in spring, a decrease was observed (of 0.5 °C) (see Fig. 15). Such a pattern of changes between temperature in historical periods (19th century and the Early Twentieth Century Arctic Warming, ETCAW) and the present was found also by Przybylak and Vizi (2005) for the Canadian Arctic; Przybylak et al. (2010) for Franz Joseph Land, Greenland and Canadian Arctic; Przybylak et al. (2022) for Greenland and Canadian Arctic; and Przybylak et al. (2024) for Greenland. The overall conclusion from this study and our earlier studies is clear: that temperature changes between historical and contemporary periods in the Arctic, particularly in the Canadian Arctic and Greenland, were evidently smallest in the warm half-year (i.e., in spring and summer) and greatest in the cold half-year. This must be taken into account in the interpretation of temperature reconstructions based on proxy data, which usually allow for temperature reconstruction of only the warm period of the year. For this reason, it is incorrect to use these reconstructions as a proxy of mean annual values, at least for the Arctic. The practice of using reconstructed summer temperatures to represent annual values is often applied in studies analysing changes in Northern Hemisphere temperature (e.g., Cook et al., 2004; Moberg et al., 2005; Ljungqvist, 2010).

Another important feature of thermal conditions (except its mean values) is temperature variability, which according to many investigations should decrease in warmer climates (e.g., Houghton et al., 1990, 1992, 1996; Karl et al., 1995, and references cited therein; Mearns et al., 1995; Zwiers and Kharin, 1998; Moberg et al., 2000; Screen, 2014). Our previous paper for SW Greenland confirms this hypothesis and the results presented here for Nain also (see Fig. 5). In all months (except August), year-to-year variability of monthly mean temperatures was greater in the colder historical period than at present (1991–2020). However, in the period 1883–1902 (not shown), the variability was greater than in the period 1771–86, which is, in turn, in opposition to that hypothesis. However, this may be related to the only minor temperature difference between these two periods, which is significantly larger compared to the modern period.

Using MDATs, it was also possible to calculate very rarely utilized climate indices (GDD, ATI, PDD and AFI) for analysis of climate change. For the late 18th century, such calculations are only available for SW Greenland (Przybylak et al., 2024). A comparison of the results from both areas (Greenland and Labrador) generally documents the same kind of changes between historical and present periods. At present, we have a longer and warmer vegetation period than in the late 18th century, and freezing periods are less frequent. These changes are in line with changes in a number of "characteristic" days. In the study period on the Labrador coast, greater/lower frequencies of all



categories of cold/warm days were observed than today. No more such information exists for the 18th century, but it is available for the 19th century and the beginning of the 20th century. Results for the Canadian Arctic (mid-19th century) presented by Przybylak and Vizi (2005) and for Novaya Zemlya (individual expedition years from 1832/33 to 1912/13) by Przybylak and Wyszyński (2017) also confirm the same tendencies.

Here, we present detailed knowledge about the climate on the Labrador coast based on a few long-term series of air temperatures (the longest and the oldest sub-daily temperature series), which are unique for this time period. The used meteorological data (quality-controlled and corrected), which are also attached to this paper for free use by any scientist, are more reliable for describing the climate and its variability in the Arctic than temperature reconstructions based on proxy data. They definitely are

of invaluable importance for historical climatology, as earlier stated by Demarée and Ogilvie (2008). The presented data should be helpful for, among other things, the identification of causes of climate change in both historical and modern times (in the latter case, mainly for estimation of anthropogenic influence in modern warming). We also hope that they can help to calibrate reconstructions based on proxy data, as well as climate simulations using climate models.


## 5 Conclusions and final remarks

The comprehensive and numerous analyses using the longest and oldest long-term (not isolated) series of air temperatures in the Arctic available for the coast of the Labrador for three sites: Nain, Okak and Hopedale, which cover the late 18th century (1771–87), allow for the formulation of the following

important conclusions about the thermal conditions and their changes to the present time:

1.   The late 18th century on the coast of Labrador, similar to the SW part of Greenland (Przybylak et al. 2024), was characterised by warm temperatures. The temperature in the period 1776-–86 was 0.7 °C higher than at the turn of the 19th century, with the greatest difference in autumn (1.3 °C) and the smallest in summer (only 0.1 °C). On the other hand, in comparison to the

contemporary period (2010–20), the temperature in Nain in the mentioned historical period was, on average, 1.4 °C lower. The same value we also obtained for Greenland (Nuuk). Very cold temperatures observed in the years 1771–73 were responsible for a significant lowering of the long-term mean temperature for the entire period available for Nain, i.e. 1771–86. In this case, the mean temperature was as much as 2.3 °C colder than today, and all months were

also colder, particularly in winter and autumn. Okak and Hopedale in historical times were also colder than in the contemporary period except from February to May. In Nain in the shorter period (1776–86), a period more comparable with Okak (1776–87) and Hopedale (1782–86), spring was also warmer than today. Analysis reveals that the majority of mean monthly air temperatures in historical times at all sites lie within two standard deviations of

the modern mean.



2.  The average monthly GDD and ATI values in historical time are usually very close to the minimum from 1991–2020, while most individual monthly values oscillate between average and minimum values from the contemporary period. The PDD during the cold season (Oct– Apr) in Nain, Okak and Hopedale were noted mainly in April and October, and their values were significantly lower than at present. On the other hand, the average monthly AFI values were higher than the present-day norm.

3.  The temperature frequency data show a trend towards more stable and less extreme temperature distributions in contemporary times compared to historical data. A greater frequency of warmer temperatures at present in comparison to the historical period is observable mainly in autumn and then in summer. On the other hand, the greatest fit between curves showing MDAT for present and historical periods occurred in spring. MDATs distribution in the historical and contemporary periods in all sites typically follows a near-normal distribution with skewness ($\gamma 1$) and kurtosis ($\gamma 2$) values not exceeding $\pm 1$.

4.  In line with expectations, all categories of distinguished cold/warm days were more/less frequent in the historical period than the contemporary period.

5.  The continentality of the Labrador climate, as well as year-to-year variability of mean monthly temperatures, were greater in historical times than at present.

Despite our efforts to obtain the best possible quality and reliability of the collected original meteorological data available from archival sources, there may still be some biases in the presented results. In our previous paper (Przybylak et al., 2024), we also analysed meteorological data gathered by the Moravian missionaries, including the existence of possible biases. That information is thus omitted here.

It is sad to note that we can probably not expect any future discovery of a series of meteorological data for the Arctic from the 18th century that is as long as those presented in the current article. However, there is still random information about weather conditions available that is described verbally in diaries (including in published missionary journals) that the Moravian brothers wrote, but from time to time also containing measured meteorological values. The importance and usefulness of such information available in Moravian missionary diaries has recently been shown by Borm et al. (2021) and previously by Demaree and Ogilvie (2008, 2011).

Finally, we hope that the corrected mean daily air temperature for the coast of Labrador from the late 18th century, accompanied by this article, together with some of our analysis, should be useful to better calibrate currently available and future temperature reconstructions based on proxy data and to help validate the climate simulations made by numerical models.



**Author contributions. Garima Singh**: Conceptualization, Methodology, Investigation, Data curation, Formal analysis, Software, Visualization, Interpretation of results, Writing – original draft, Writing – review & editing. **Rajmund Przybylak**: Conceptualization, Methodology, Investigation, Data collection and selection, Formal analysis, Funding acquisition, Project administration, Validation, Supervision, Interpretation of results, Writing – original draft, Writing – review & editing. **Przemysław Wyszyński**: Conceptualization, Data collection and selection, Software, Visualisation, Validation. **Andrzej Araźny:** Conceptualization, Data collection and selection, Validation. **Konrad Chmist:** Data curation, Visualisation.

**Declaration of Competing Interest.** The authors declare that they have no known competing financial interests or personal relationships that could have appeared to influence the work reported in this paper.

**Financial support.** This research has been supported by the Narodowe Centrum Nauki (grant no. 2020/39/B/ST10/00653

**Data availability**. Datasets for this research were derived from the following public domain resources:

1. Historical meteorological data for Nain, Hopedale and Okak in the late 18th century (1771–87): Repository for Open Data (RepOD), Nicolaus Copernicus University Centre for Climate Change Research collection, https://doi.org/10.18150/VJJFOE, as cited in Singh et al. (2024),

 2. Present observational meteorological data for Nain and Hopedale stations in 1990 – 2020: Canadian Centre for Climate Services, https://climate.weather.gc.ca/historical_data/search_historic_data_e.html

 3. ERA5 reanalysis data, nearest grid point to Okak, period 1990-2020: https://cds.climate.copernicus.eu/cdsapp#!/dataset/reanalysis-era5-complete?tab=overview



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
