# Peer review of "Thermal conditions on the coast of Labrador during the late 18th century"

_EGUsphere, 2024_

## Author Response (AR1)

Dear Editor,

We would like to thank you and the anonymous reviewer, as well as Gaston Demarée (the second reviewer), for providing positive feedback and constructive comments about our manuscript. We believe that these comments helped us improve the description of our work.

**Reviewer 1**

The comparison with proxy data (dendrochronology) comes a bit abruptly in the results chapter, it should perhaps have been mentioned in the methods/data chapter beforehand. Otherwise the discussion must be discussed in a different way, e.g. excluding proxy data. But I think the first suggestion can help.

Reply: We would like to thank you for evaluating our work and for your valuable feedback and comments. However, it is difficult to answer your question/suggestion because in the results chapter, we do not present any results based on dendrochronological data. There must be some mistake here. Only in the Discussion part did we, for comparison purposes, use a number of reconstructions of air temperature based on different proxy data (not only dendrochronological data). This is a necessary condition for conducting a comparative analysis of the results obtained in this article with other results from the same period in the absence of instrumental measurements from the immediate vicinity.

 References to wikipedia should not be used for definitions.

Reply: Thank you for your suggestion. We have provided another reference documenting this fact (Black, H.D., Gebel, G., Newton, R.R.: The centenary of the Prime Meridian and of International Standard Time, John Hopkins APL Technical Digest 5(4), 381-389, 1984).

It is positive that rare data is included in scientific forums.

Reply: Thank you for this positive estimate of our efforts in data rescue of old meteorological measurements.

It is not as consistent what is used in the comparison (reanalysis or dendrochronology)

Reply: Thank you for this comment. However, this comment is very general and does not contain a reference to the specific text (lines in the manuscript), so it is very difficult to guess what the statement means. We want to say that data from the ERA5 reanalysis were only used to reconstruct present data for Okak (the meteorological station does not exist now here) and to fill a small number of gaps in meteorological data available for Nain and Hopedale. We can also inform you that data taken from the reanalysis for Okak were corrected using a comparison of instrumental observations from Nain and data from ERA5 reanalysis for the nearest grid. For more details, see lines 189-197

**Reviewer 2**

Lines 165-180: Figure 2. Examples of manuscripts presenting meteorological observations. The manuscripts of meteorological observations by the Moravian Brethren are from 3 archives: Unitätsarchiv at Herrnhut, the Moravian Archives at Muswell Hill, London, and the Archives of the Royal Society, London. Although 4 manuscript pages are shown in figure 2, none of the pictures represent a manuscript page from the Moravian Archives at Muswell, London. By the way, there is a small editing error in the word: Unitätsarchiv in the legend of Figure 2.

Reply: Thank you for this comment. We corrected the error in the word „Unitätsarchiv".

Indeed, there is no example of the source from the Muswell Hills archive in Fig. 2, but sources from this archive were used in our work also, for Okak (1776-1787). They were even longer than those available in the Royal Society archive in London (1779-84). For greater clarity, we modified our text a little; see the text below:

All these records were sourced from three primary archival collections: i) The Moravian Archives in Herrnhut (Germany) and the Royal Society in London (United Kingdom), ii) The Moravian Archives at Muswell Hill (United Kingdom) and the Royal Society in London (United Kingdom), the second archive has data only for years 1779-84, and iii) the archives of the Royal Society in London (United Kingdom), respectively. Examples of some used sources are shown in Fig. 2 (excluding sources from the Moravian Archives at Muswell Hill), while their data coverage is shown in Fig. 3.

Lines 200-215: The comparison of the different formulas for the mean daily air temperature MDAT [1] - [10] is highly interesting. The meteorological observation times at the Moravian missionary stations Nain, Hopedale and Okak occurred in a time when ideas on the mean daily air temperature were not yet well established. One has to wait until the mid of the 19th century when clear ideas about the mean daily air temperature were taking shape. The mean idea to define a mean daily air temperature was to compare temperature worldwide. However, one had to wait until the establishment of the International Meteorological Organization (IMO) in the late 19th and early 20th centuries to define a standard mean daily air temperature. This implies that the WMO standard mean daily air temperature might be slightly different from the ones computed in the present manuscript. Most probably, it is of the order of the range in the monthly corrections values in Table 2. Nevertheless, it is also possible to estimate the WMO standard mean daily air temperature of the historical meteorological stations in Labrador by linear regression techniques using the contemporaneous meteorological data at the same stations.

Reply: Thank you for this comment. In the text, we wrote, perhaps not very clearly, that we (and not Moravian observers) calculated the daily averages using the formulas mentioned in the manuscript. Throughout the history of meteorological and climatic research, various methods have been used to calculate the daily average; however, with the introduction of automated meteorological observations, the average of 24 measurements at hourly intervals is recommended as the best daily average.

Therefore, in order to be able to reliably compare daily averages calculated based on several measurements per day, it is necessary to introduce corrections in relation to the 24-hour average. Such a procedure was utilised in our paper.

Line 218: I have problems with the appearance of T3 in the formula MDAT4 as T3 doesn't appear in the Moravian observation schemes.

Reply: We would like to inform you that meteorological observations at 3 a.m. were conducted very briefly in Nain between July 1 and 24, 1774, and we used them in our work.

Lines 690-695: the full reference of the paper is as follows:

Demarée, G., Ogilvie, A.E.J. & Mailier, P. (2020) Early meteorological observations in Greenland and Labrador in the 18th century: a contribution of the Moravian Brethren. Proceedings of the 35th International Symposium on the Okhotsk Sea & Polar Oceans 2020, Mombetsu-2020 Symposium, 16-19 February 2020, Okhotsk Sea and Polar Oceans Research Association (OSPORA), Mombetsu, Hokkaido, Japan, p. 35-38.

Reply: Thank you very much. The reference was corrected.

Lines 695-700: the book edited by Felicity Jensz and Christina Petterson in 2021 has been recently translated in German language and the contribution by Demarée and Ogilvie appears as follows:

Demarée, G. R. und Ogilvie, A. E.J. (2024) Frühe meteorologische Beobachtungen in Grönland: Die Beiträge von David Cranz, Christian Gottlieb Kratzenstein und Christopher Brasen. In : Felicity Jensz u. Christina Petterson (Hrsg.) Vermächtnisse von David Cranz' „Historie von Grönland" (1765). Springer, p. 149-172.

Reply: Thank you for this information. We added this reference to the text.

General conclusion

The manuscript is an excellent contribution to the study of the historical climate of the Labrador Coast at the end of the 18th century. The paper compares also the historical climate of the end of the 18th century with two sets of contemporaneous climate data. This comparison is based upon the definitions of the mean daily air temperature [1] - [10]. In this context, the manuscript is fully novel and is a fundamental contribution to the knowledge of the climate in the Arctic at the end of the 18th century.

Reply: Thank you very much.

The analysis of comparison of historical meteorological data at 3 Moravian missionary stations at the end of the 18th century with contemporaneous meteorological data at the same stations is carried out on monthly, seasonal and annual basis. Similarly, specialized climate indices (after Øyvind Nordli et al., 2020) were computed using the corrected MDAT values.

Reply: Thank you very much.

This Reviewer suggests accepting publication of the manuscript as it stands taking care of the remarks mentioned in this review.

Thank you very much for the very helpful comments and suggestions.

The authors have fully developed the comparison of the historical and contemporaneous meteorological observations on the Coast of Labrador. This manuscript is partly built upon a catalogue of centuries-long meteorological observations by the missionaries of the Moravian Brethren formerly published. At this stage, historical climatologists remain fully interested in the voice of the Moravian missionaries through random information available from their diaries

commenting the climate conditions and remarkable events such as the dry fog of the year 1783, earthquakes, forest fires, etc.

Reply: We would like to thank you for evaluating our work and for your valuable feedback and comments.